# Evaluation Metrics for Reading Comprehension: Prerequisite Skills and Readability

## Abstract

Knowing the quality of reading comprehension (RC) datasets is important for the development of natural language understanding systems. In this study, two classes of metrics were adopted for evaluating RC datasets: prerequisite skills and readability. We applied these classes to six existing datasets, including MCTest and SQuAD, and demonstrated the characteristics of the datasets according to each metric and the correlation between the two classes. Our dataset analysis suggested that the readability of RC datasets does not directly affect the question difficulty and that it is possible to create an RC dataset that is easy-to-read but difficult-to-answer.

## 1 Introduction

A major goal of natural language processing (NLP) is to develop agents that can understand natural language. Such an ability can be tested with a reading comprehension (RC) task that requires the agent to read open-domain documents and answer questions about them. Building the RC ability is challenging because RC comprises multiple processes including parsing, understanding cohesion, and inference with linguistic and general knowledge.

Clarifying what a system achieves is important to the development of RC systems. To achieve robust improvement, systems need to be measured according to various metrics, not just simple accuracy. However, a current problem is that most RC datasets are presented only with superficial categories, such as question types (e.g., what, where, and who) and answer types (e.g., numeric, location, and person). In addition, Chen et al. (2016) revealed that some questions in datasets may not

---

**ID:** SQuAD, United_Methodist_Church
**Context:** The United Methodist Church (UMC) practices infant and adult baptism. Baptized Members are those who have been baptized as an infant or child, but who have not subsequently professed their own faith.
**Question:** What are members who have been baptized as an infant or child but who have not subsequently professed their own faith?
**Answer:** Baptized Members

---

**ID:** MCTest, mc160.dev.8
**Context:** Sara wanted to play on a baseball team. She had never tried to swing a bat and hit a baseball before. Her Dad gave her a bat and together they went to the park to practice.
**Question:** Why was Sara practicing?
**Answer:** She wanted to play on a team

---

Figure 1: Examples of RC questions from SQuAD (Rajpurkar et al., 2016) and MCTest (Richardson et al., 2013) (the contexts are excerpted).

have the quality to test RC systems. In these situations, it is difficult to assess systems accurately.

As Norvig (1989) stated, questions easy for humans often turn out to be difficult for systems. For example, see the two RC questions in Figure 1. In the first example from SQuAD (Rajpurkar et al., 2016), although the document is taken from a Wikipedia article and is thus written for adults, the question is easy to solve simply look at one sentence and select an entity. On the other hand, in the second example from MCTest (Richardson et al., 2013), the document is written for children and is easy to read, but the question requires reading multiple sentences and a combination of several skills, such as understanding of causal relations (*Sara wanted...* → *they went to...*), coreference resolution (*Sara* and *Her Dad* = *they*), and complementing ellipsis (*baseball team = team*). These two examples show that the readability of the text does not necessarily correlate with the difficulty of the questions. Nevertheless, the accompanying categories of existing RC datasets cannot

provide any way to analyze this issue.

In this study, our goal was to investigate how the two difficulties of "answering questions" and "reading text" relate in RC. Therefore, we adopted two classes of evaluation metrics for RC datasets to analyze both the quality of datasets and the performance of systems. Our paper is divided into the following sections. First, we adopt the two classes: *prerequisite skills* and *readability* (Section 3). We then specify evaluation metrics for each (Sections 3.1 and 3.2). Next, we annotate six existing RC datasets with these metrics (Section 4). Finally, we present our annotation results (Section 5) and discuss them (Section 6).

Our two classes of metrics are based on McNamara and Magliano (2009)'s analysis of human text comprehension in psychology. The first class defines the difficulty of comprehending the context to answer questions. We adopted prerequisite skills proposed in a previous study by Sugawara et al. (2017). That study presented an important observation of the relation between the difficulty of an RC task and prerequisite skills: the more skills that are required to answer a question, the more difficult the question is. From this observation, we assume that the number of required skills corresponds to the difficulty of a question. This is because each skill corresponds to a function of a system, which has to be equipped with the system. However, a problem in previous studies, including that of Sugawara and Aizawa (2016), is that they analyzed only two datasets and that their categorization of knowledge reasoning is provisional with a weak theoretical background.

Therefore, we reorganized the category of knowledge reasoning in terms of textual entailment and human text comprehension. In research on textual entailment, several methodologies have been proposed for precise analysis of entailment phenomena (Dagan et al., 2013; LoBue and Yates, 2011). In psychology research, Kintsch (1993) proposed dichotomies for the classification of human inferences: *retrieved* versus *generated*. In addition, McNamara and Magliano (2009) proposed a similar distinction for inferences: *bridging* versus *elaboration*. We utilized these insights in order to develop a comprehensive but not overly specific classification of knowledge reasoning.

The second class defines the difficulty of reading contents, readability, in documents considering syntactic and lexical complexity. We leverage a wide range of linguistic features proposed by Vajjala and Meurers (2012).

In the annotation, annotators selected sentences needed for answering and then annotated them with prerequisite skills under the same condition for RC datasets with different task formulations. Therefore, our annotation was equivalent in that the datasets were annotated from the point of view of whether a context entails a hypothesis that can be made from a question and its answer. This means our methodology could not evaluate the competence of looking for sentences that need to be read and answer candidates from the context. Therefore, our methodology was used to evaluate the understanding of contextual entailments in a broader sense for RC.

The contributions of this paper are as follows:

1. We adopt two classes of evaluation metrics to show the qualitative features of RC datasets. Through analyses of RC datasets, we demonstrate that there is only a weak correlation between the difficulty of questions and the readability of context texts in RC datasets.

2. We revise the previous classification of prerequisite skills for RC. Specifically, skills of knowledge reasoning are organized in terms of entailment phenomena and human text comprehension in psychology.

3. We annotate six existing RC datasets with our organized metrics for the comparison and make the results publicly available.

We believe that our annotation results will help researchers to develop a method for the step-by-step construction of better RC datasets and a method to improve RC systems.

## 2 Related Work

### 2.1 Reading Comprehension Datasets

In this section, we present a short chronicle of RC datasets. To our knowledge, Hirschman et al. (1999) were the first to use NLP methods for RC. Their dataset consisted of reading materials for grades 3–6 with simple 5W (*wh-*) questions. Afterwards, investigations into natural language understanding questions mainly focused on other formulations, such as question answering (Yang et al., 2015; Wang et al., 2007; Voorhees et al., 1999) and textual entailment (Bentivogli et al., 2010; Sammons et al., 2010; Dagan et al., 2006).

One of the RC tasks following it was QA4MRE (Sutcliffe et al., 2013). The top accuracy of this task remained 59% at that time, and the size of the dataset was very limited: there were only 224 gold-standard questions, which is not enough for machine learning methods.

Thus, an important issue for designing RC datasets is their scalability. Richardson et al. (2013) presented MCTest, which is an open-domain narrative dataset for gauging comprehension at a childs level. This dataset was created by crowdsourcing and is based on scalable methodology. Since then, more large-scale datasets have been proposed with the development of machine learning. For example, the CNN/Daily Mail dataset (Hermann et al., 2015) and CBTest (Hill et al., 2016) have approximately 1.4M and 688K passages, respectively. These context texts and questions were automatically curated and generated from large corpora. However, Chen et al. (2016) indicated that approximately 25% of the questions in the CNN/Daily Mail dataset are unsolvable or nonsense. This has highlighted the demand for more stable and robust sourcing methods regarding dataset quality.

In addition to this, several RC datasets were presented in the last half of 2016 with large documents and sensible queries that were guaranteed by crowdsourcing or human testing. We explain those datasets in Section 4.2. They were aimed at achieving large and good-quality content for machine learning models. Nonetheless, as shown in the examples of Figure 1, there is room for improvement, and there is still no methodology available for evaluating the quality.

## 2.2 Reading Comprehension in Psychology

In psychology, there is a rich tradition of research on human text comprehension. The construction–integration (C–I) model (Kintsch, 1988) is one of the most basic and influential theories. This model assumes connectional and computational architecture for text comprehension. It assumes that comprehension is the processing of information based on the following steps:[1]

1. *Construction:* Read sentences or clauses as inputs; form and elaborate concepts and propositions corresponding to the inputs.

2. *Integration:* Associate the contents to consistently understand them (e.g., coreference,

---

[1] Note that this is only a very simplified overview.

discourse, and coherence).

During these steps, three levels of representation are constructed (Van Dijk and Kintsch, 1983): the *surface code* (i.e., wording and syntax), the *textbase* (i.e., text propositions with cohesion), and the *situation model* (i.e., mental representation). Based on the above assumptions, McNamara and Magliano (2009) proposed two aspects of text comprehension: "strategic/skilled comprehension" and "text ease of processing." We employed these assumptions as the basis of two classes of evaluation metrics (Section 3).

On the other hand, Kintsch (1993) proposed two dichotomies for the classification of human inferences, including the knowledge-based ones that are performed in the C–I model. The first is whether inferences are *automatic* or *controlled*. However, Graesser et al. (1994) indicated that this distinction is ambiguous because there is a continuum between the two states that depends on individuals. Therefore, this dichotomy is not suited for empirical evaluation, which we are working on attempting. The second is whether inferences are *retrieved* or *generated*. *Retrieved* means that the information used for inference is retrieved from context. In contrast, when inferences are *generated*, the reader uses external knowledge going beyond the context.

A similar distinction was proposed by McNamara and Magliano (2009): *bridging* and *elaboration*. Bridging inference connects current information to other information that was previously encountered. Elaboration connects current information to external knowledge that is not in a context. We use these two types of inferences in the classification of knowledge reasoning.

## 3 Evaluation Metrics for Datasets

Based on McNamara and Magliano (2009)'s depiction of text comprehension, we adopted two classes for the evaluation of RC datasets: *prerequisite skills* and *readability*.

We refined the prerequisite skills (Section 3.1) for RC that were proposed by Sugawara et al. (2017) and Sugawara and Aizawa (2016) using the insights mentioned in the previous section. This class covers the *textbase* and *situation model*, or understanding each fact and associating multiple facts in a text: relations of events, characters, topic of story, and so on. This class also includes knowledge reasoning; this is divided into several met-

rics according to the distinctions of human inferences, as discussed by Kintsch (1993) and McNamara and Magliano (2009), and according to the classification of entailment phenomena by Dagan et al. (2013) and LoBue and Yates (2011).

Readability metrics (Section 3.2) intuitively describe the difficulty of reading: vocabulary and the complexity of texts, or the *surface code*.

## 3.1 Prerequisite Skills

By refining Sugawara et al. (2017)'s ten reading comprehension skills, we reorganized thirteen prerequisite skills in total, which are presented below. Skills that have been modified/elaborated from the original definition are appended with an asterisk (*), and new skills unique to this study are appended with a dagger ([†]).

**1. Object tracking**[*]: Jointly tracking or grasping of multiple objects, including set or membership (Clark, 1975). This skill is a renamed version of *list/enumeration* used in the original classification to emphasize its scope for multiple objects.

**2. Mathematical reasoning**[*]: We merged statistical and quantitative reasoning with mathematical reasoning. This skill is a renamed version of *mathematical operations*.

**3. Coreference resolution**[*]: This skill has a small modification: it includes one anaphora (Dagan et al., 2013). This skill is similar to *direct reference* (Clark, 1975).

**4. Logical reasoning**[*]: We reorganized this skill as understanding predicate logic, e.g., conditionals, quantifiers, negation, and transitivity. Note that *mathematical reasoning* and this skill are intended to align with offline skills mentioned by Graesser et al. (1994).

**5. Analogy**[*]: Understanding of metaphors including metonymy and synecdoche. See LoBue and Yates (2011) for examples of synecdoche.

**6. Causal relation:** Understanding of causality that is represented by explicit expressions of why, because, the reason... (only if they exist).

**7. Spatiotemporal relation:** Understanding of spatial and/or temporal relationships between multiple entities, events, and states.

For commonsense reasoning in the original classification, we defined the following four categories, which can be jointly required.

**8. Ellipsis**[†]: Recognizing implicit/omitted information (argument, predicate, quantifier, time, place). This skill is inspired by Dagan et al. (2013)

and the discussion of Sugawara et al. (2017).

**9. Bridging**[†]: Inferences between two facts supported by grammatical and lexical knowledge (e.g., synonymy, hypernymy, thematic role, part of events, idioms, and apposition). This skill is inspired by the concept of *indirect reference* in the literature (Clark, 1975). Note that we excluded *direct reference* because it is *coreference resolution* (pronominalization) or *elaboration* (epithets).

**10. Elaboration**[†]: Inference using known facts, general knowledge (e.g., kinship, exchange, typical event sequence, and naming), and implicit relations (e.g., noun-compound and possessive) (see Dagan et al. (2013) for details). *Bridging* and *elaboration* are distinguished by whether knowledge used in inferences is grammatical/lexical versus general/commonsense.

**11. Meta-knowledge**[†]: Using knowledge including a reader, writer, and text genre (e.g., narratives and expository documents) from meta-viewpoints (e.g., *Who are the principal characters of the story?* and *What is the main subject of this article?*). Although this skill can be regarded as part of *elaboration*, we defined an independent skill because this knowledge is specific to RC.

The last two skills are intended to be performed on a single sentence.

**12. Schematic clause relation**: Understanding of complex sentences that have coordination or subordination, including relative clauses.

**13. Punctuation**[*]: Understanding of punctuation marks (e.g., parenthesis, dash, quotation, colon, and semicolon). This skill is a renamed version of *special sentence structure*. Concerning the original definition, we regarded "scheme" in figures of speech as ambiguous and excluded it (*ellipsis* was defined as a skill, and apposition was merged into *bridging*). We did the same with understanding of constructions, which was merged into idioms in *bridging*.

Note that we did not construct this classification to be dependent on RC models. This is because our methodology is intended to be general and applicable to many kinds of architectures.

## 3.2 Readability Metrics

In this study, we evaluated the readability of texts based on metrics in NLP. Several studies have examined readability for various applications, such

---

[4] http://en.wikipedia.org/wiki/Academic_Word_List

- Ave. num. of characters per word (*NumChar*)
- Ave. num. of syllables per word (*NumSyll*)
- Ave. sentence length in words (*MLS*)
- Proportion of words in AWL (*AWL*)
- Modifier variation (*ModVar*)
- Num. of coordinate phrases per sentence (*CoOrd*)
- Coleman–Liau index (*Coleman*)
- Dependent clause to clause ratio (*DC/C*)
- Complex nominals per clause (*CN/C*)
- Adverb variation (*AdvVar*)

Figure 2: Readability metrics. *AWL* refers to the Academic Word List.[4]

as second language learning (Razon and Barnden, 2015) and text simplification (Aluisio et al., 2010), and various aspects, such as development measures of second language acquisition (Vajjala and Meurers, 2012) and discourse relations (Pitler and Nenkova, 2008).

Of these, we used the classification of linguistic features proposed by Vajjala and Meurers (2012). This is because they presented a comparison of a wide range of linguistic features focusing on second language acquisition and their method can be applied to plain text.[2]

We list the readability metrics in Figure 2, which were reported by Vajjala and Meurers (2012) as the top ten features that affect human readability. To classify these metrics, we can place three classes: lexical features (*NumChar*, *NumSyll*, *AWL*, *AdvVar*, and *ModVar*), syntactic features (*MLS*, *CoOrd*, *DC/C*, and *CN/C*), and traditional features (*Coleman*). We applied these metrics only to sentences that needed to be read to answer questions. Nonetheless, because these metrics were proposed for human readability, they do not necessarily correlate with those for RC systems. Therefore, in a system analysis, we ideally have to consult various features.[3]

## 4 Annotation of Reading Comprehension Datasets

We annotated six existing RC datasets with the prerequisite skills. We explain the annotation procedure in Section 4.1 and the specifications of the annotated RC datasets in Section 4.2.

---

[2]Pitler and Nenkova (2008)'s work is more suitable for measuring text quality. However, we could not use their results because we did not have discourse annotations.

[3]We will make available the analysis of RC datasets on basic features as much as possible.

| RC dataset | Genre | Query sourcing | Task formulation |
|---|---|---|---|
| QA4MRE | Technical documents | Handcrafted by experts | Multiple choice |
| MCTest | Narratives by crowd workers | Crowdsourced | Multiple choice |
| SQuAD | Wikipedia articles | Crowdsourced | Text span selection |
| Who-did-What | News articles (Gigaward v5) | Automated from other articles | Cloze |
| MS MARCO | Segmented web pages | Search engine queries | Description |
| NewsQA | News articles | Crowd sourced | Text span selection |

Table 1: Analyzed RC datasets, their genres, query sourcing methods, and task formulations.

### 4.1 Annotation Procedure

We asked three annotators to annotate questions of RC datasets with the prerequisite skills that are required to answer each question. We allowed multiple labeling. For each task that was curated from the datasets, the annotators saw the context, question, and its answer jointly. When a dataset consisted of multiple choice questions, we showed all candidates and labeled the correct one with an asterisk. The annotators then selected sentences that needed to be read to answer the question and decided if each prerequisite skill was required. The annotators were allowed to select *nonsense* for an unsolvable question to distinguish it from a solvable question that required no skills.

### 4.2 Dataset Specifications

For the annotation, we randomly selected 100 questions from each dataset in Table 1. This amount of questions was considered to be sufficient for the analysis of RC datasets as performed by Chen et al. (2016). The questions were sampled from the gold-standard dataset of QA4MRE and the development sets of the other RC datasets. We explain the method of choosing questions for the annotation in Appendix A.

There were other datasets we did not annotate in this study. We decided not to annotate those datasets because of the following reasons. CNN/Daily Mail (Hermann et al., 2015) is anonymized and contains some errors (Chen et al., 2016), so it did not seem to be suitable for annotation. We considered CBTest (Hill et al., 2016) to be for language modeling tasks rather than RC task and excluded it. LAMBADA (Paperno et al., 2016)'s texts are formatted for machine reading,

| Skills | QA4MRE | MCTest | SQuAD | WDW | MARCO | NewsQA |
|---|---|---|---|---|---|---|
| 1. Tracking | 11.0 | 6.0 | 3.0 | 8.0 | 6.0 | 2.0 |
| 2. Math. | 4.0 | 4.0 | 0.0 | 3.0 | 0.0 | 1.0 |
| 3. Coref. resol. | 32.0 | 49.0 | 13.0 | 19.0 | 15.0 | 24.0 |
| 4. Logical rsng. | 15.0 | 2.0 | 0.0 | 8.0 | 1.0 | 2.0 |
| 5. Analogy | 7.0 | 0.0 | 0.0 | 7.0 | 0.0 | 3.0 |
| 6. Causal rel. | 1.0 | 6.0 | 0.0 | 2.0 | 0.0 | 4.0 |
| 7. Sptemp rel. | 26.0 | 9.0 | 2.0 | 2.0 | 0.0 | 3.0 |
| 8. Ellipsis | 13.0 | 4.0 | 3.0 | 16.0 | 2.0 | 15.0 |
| 9. Bridging | 69.0 | 26.0 | 42.0 | 59.0 | 36.0 | 50.0 |
| 10. Elaboration | 60.0 | 8.0 | 13.0 | 57.0 | 18.0 | 36.0 |
| 11. Meta | 1.0 | 1.0 | 0.0 | 0.0 | 0.0 | 0.0 |
| 12. Clause rel. | 52.0 | 40.0 | 28.0 | 42.0 | 27.0 | 34.0 |
| 13. Punctuation | 34.0 | 1.0 | 24.0 | 20.0 | 14.0 | 25.0 |
| Nonsense | 10.0 | 1.0 | 3.0 | 27.0 | 14.0 | 1.0 |

Table 2: Frequencies (%) of prerequisite skills needed for the RC datasets.

| #Skills | QA4MRE | MCTest | SQuAD | WDW | MARCO | NewsQA |
|---|---|---|---|---|---|---|
| 0 | 2.0 | 18.0 | 27.0 | 2.0 | 15.0 | 13.0 |
| 1 | 13.0 | 36.0 | 33.0 | 5.0 | 35.0 | 26.0 |
| 2 | 13.0 | 24.0 | 24.0 | 14.0 | 29.0 | 23.0 |
| 3 | 20.0 | 15.0 | 6.0 | 22.0 | 6.0 | 25.0 |
| 4 | 14.0 | 4.0 | 6.0 | 16.0 | 2.0 | 9.0 |
| 5 | 13.0 | 1.0 | 1.0 | 6.0 | 0.0 | 2.0 |
| 6 | 10.0 | 1.0 | 0.0 | 6.0 | 0.0 | 1.0 |
| 7 | 1.0 | 0.0 | 0.0 | 2.0 | 0.0 | 0.0 |
| 8 | 1.0 | 0.0 | 0.0 | 0.0 | 0.0 | 0.0 |
| 9 | 0.0 | 0.0 | 0.0 | 0.0 | 0.0 | 0.0 |
| 10 | 3.0 | 0.0 | 0.0 | 0.0 | 0.0 | 0.0 |
| Ave. | 3.25 | 1.56 | 1.28 | 2.43 | 1.19 | 1.99 |

Table 3: Frequencies (%) of the number of required prerequisite skills for the RC datasets.

and all tokens are lowercased, which seemingly prevents inferences based on proper nouns. Thus, we decided that its texts are not suitable for human reading and annotation.

## 5   Results of the Dataset Analysis

We present the results from evaluating the RC datasets according to the two classes of metrics. The inter-annotator agreement was 90.1% for 62 randomly sampled questions. The evaluation was performed according to the following four points of view (i)–(iv).

(i) **Frequencies of prerequisite skills** (Table 2): QA4MRE had the highest scores for frequencies among the datasets. This seems to reflect the fact that QA4MRE has technical documents that contain a wide range of knowledge (*bridging* and *elaboration*), multiple clauses, and punctuation and that the questions are devised by experts.

MCTest achieved high scores in several skills (first in *causal relation* and *meta-knowledge* and second in *coreference resolution* and *spatiotemporal relation*) and lower score in *punctuation*. These scores seem to be because the MCTest dataset consists of narratives.

| Metrics | QA4MRE | MCTest | SQuAD | WDW | MARCO | NewsQA |
|---|---|---|---|---|---|---|
| NumChar | 5.026 | *3.892* | **5.378** | 4.988 | 5.016 | 5.017 |
| NumSyll | 1.663 | *1.250* | **1.791** | 1.657 | 1.698 | 1.635 |
| MLS | 28.488 | *11.858* | 23.479 | **29.146** | 19.634 | 22.933 |
| AWL | 0.067 | *0.003* | **0.071** | 0.033 | 0.047 | 0.038 |
| ModVar | 0.174 | *0.114* | **0.188** | 0.150 | 0.186 | 0.138 |
| CoOrd | **0.922** | *0.309* | 0.722 | 0.467 | 0.651 | 0.507 |
| Coleman | 12.553 | *4.333* | **14.095** | 12.398 | 11.836 | 12.138 |
| DC/C | **0.343** | 0.223 | 0.243 | 0.254 | *0.220* | 0.264 |
| CN/C | 1.948 | *0.614* | 1.887 | **2.310** | 1.935 | 1.702 |
| AdvVar | **0.038** | 0.035 | 0.032 | *0.019* | 0.022 | *0.019* |
| Flesch | 14.953 | *3.607* | 14.678 | **15.304** | 12.065 | 12.624 |
| Words | **1545.7** | 174.1 | 130.4 | 253.7 | *70.7* | 638.4 |

Table 4: Results of readability metrics for the RC datasets. *Flesch* represents the Flesch–Kincaid grade level, which we calculated as an intuitive reference for the readability. This value represents the education level required to understand the text. *Words* means the average word count of the context for each question.

Another dataset that achieved remarkable scores is Who-did-What. This dataset achieved the highest score for *ellipsis*. This is because the questions of Who-did-What are automatically generated from articles not used as context. This methodology can avoid textual overlap between a question and its context; therefore, the skills of *ellipsis*, *bridging*, and *elaboration* are frequently required.

With regard to *nonsense*, MS MARCO and Who-did-What received relatively high scores. This appears to have been caused by the automated curation, which may generate separation between the contents of the context and question (i.e., web segments and a search query in MS MARCO, and a context article and question article in Who-did-What). In stark contrast, NewsQA had no nonsense questions. Although this result was affected by our filtering described in Appendix A, it is important to note that the NewsQA dataset includes annotations of meta information whether or not a question makes sense (*is_question_bad*).

(ii) **Number of required prerequisite skills** (Table 3): QA4MRE had the highest score; on average, each question required 4.6 skills. There were few questions in QA4MRE that required zero or one skill, whereas the other datasets contained some questions that required zero or one skill. Table 3 also indicates that more than 90% of the MS MARCO questions required fewer than three skills, at least according to the annotation.

(iii) **Readability metrics for each dataset** (Table 4): SQuAD and QA4MRE achieved the highest scores in most metrics; this reflects the fact that

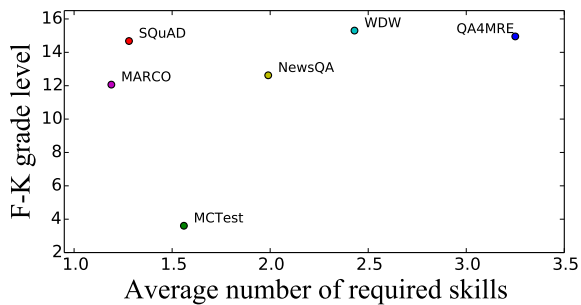

Figure 3: Flesch–Kincaid grade levels and average number of required prerequisite skills for the RC datasets.

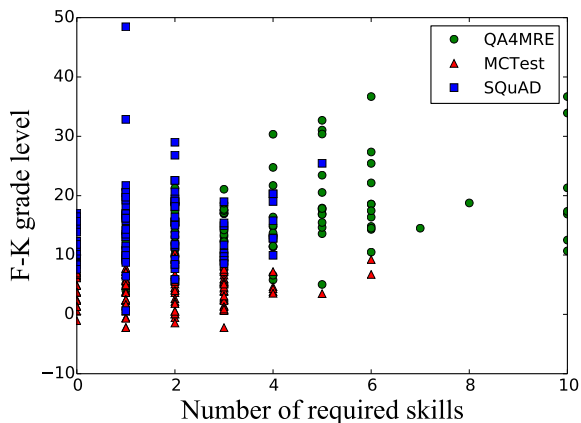

Figure 4: Flesch–Kincaid grade levels and number of required prerequisite skills for all questions of the selected RC datasets.

Wikipedia articles and technical documents generally require a high grade level to understand. In contrast, MCTest had the lowest scores; its dataset consist of narratives for children.

**(iv) Correlation between numbers of required prerequisite skills and readability metrics** (Figure 3, Figure 4, and Table 5): Our main interest was in the correlation between prerequisite skills and readability. To investigate this, we examined the relation between the number of required prerequisite skills and readability metrics (represented by the Flesch–Kincaid grade level), as shown in Figure 3 for each dataset and in Figure 4 for each question. The first figure shows the trends of the datasets. QA4MRE was relatively difficult both to read and to answer, and SQuAD was difficult to read but easy to answer. For further investigation, we selected three datasets (QA4MRE, MCTest, and SQuAD) and plotted all of their questions in the second figure. Three separate domains can be seen.

Table 5 presents Pearson's correlation coeffi-

| Metrics | $r$ | $p$ | Metrics | $r$ | $p$ |
|---------|-----|-----|---------|-----|-----|
| NumChar | 0.068 | 0.095 | CoOrd | 0.166 | 0.000 |
| NumSyll | 0.057 | 0.161 | Coleman | 0.140 | 0.001 |
| MLS | 0.416 | 0.000 | DC/C | 0.188 | 0.000 |
| AWL | 0.114 | 0.005 | CN/C | 0.131 | 0.001 |
| ModVar | 0.025 | 0.545 | AdvVar | 0.026 | 0.515 |
| Flesch | 0.343 | 0.000 | Words | 0.355 | 0.000 |

Table 5: Pearson's correlation coefficients ($r$) with the p-values ($p$) of the readability metrics and number of required prerequisite skills for all questions of the RC datasets.

cients between the number of required prerequisite skills and each readability metric for all questions of the RC datasets. Although there are weak correlations from 0.025 to 0.416, the results highlight that there is not necessarily a strong correlation between the two values. This leads to the following two insights. First, the readability of RC datasets does not directly affect the difficulty of their questions. That is, RC datasets that are difficult to read are not necessary difficult to answer. Second, it is possible to create difficult questions from context that is easy to read. MCTest is a good example. The context texts of MCTest dataset are easy to read, but the difficulty of the questions is comparable to that of the other datasets.

To summarize our results in terms of each RC dataset, we present the following observations:

- **QA4MRE** seemed to be the most difficult dataset among the RC datasets we analyzed, whereas MS MARCO seemed to be the easiest.
- **MCTest** is a good example of an RC dataset that is easy to read but difficult to answer. The corpus' genre (i.e., narrative) seems to reflect the trend of required skills for the questions.
- **SQuAD** was difficult to read along with QA4MRE but relatively easy to answer compared to other datasets.
- **Who-did-What** performed well in terms of its query sourcing method. Although its questions are automatically created, they are sophisticated in terms of knowledge reasoning. However, an issue with the automated sourcing method is excluding nonsense questions.
- **MS MARCO** was a relatively easy dataset in terms of prerequisite skills. A problem is that the dataset contained nonsense questions.
- **NewsQA** is advantageous in that it provides meta information on the reliability of the ques-

tions. Such information enabled us to avoid using nonsense questions, such as in the training of machine learning models.

## 6 Discussion

In this section, we discuss several matters regarding the construction of RC datasets and the development of RC systems using our methodology.

**How to utilize the two classes of metrics for system development**: One example for the development of an RC system is that it should be built to solve an easy-to-read and easy-to-answer dataset. The next step is to improve the system so that it can solve an easy-to-read and difficult-to-answer dataset. Finally, only after it can solve such dataset should the system be applied to a difficult-to-read and difficult-to-answer dataset. Appropriate datasets can be prepared for every step by measuring their properties using the metrics of this study. Such datasets can be placed in a continuum based on the grades of the metrics and applied to each step of the development, like in curriculum learning (Bengio et al., 2009) and transfer learning (Pan and Yang, 2010).

**Corpus genre**: Attention should be paid to the genre of corpus used to construct a dataset. Expository documents like news articles tend to require factorial understanding. Most existing RC datasets use such texts because of their availability. On the other hand, narrative texts have a close correspondence to our everyday experience, such as the emotion and intention of characters (Graesser et al., 1994). If we want to build agents that work in the real world, RC datasets may have to be constructed from narratives.

**Question type**: In contrast to factorial understanding, comprehensive understanding of natural language texts needs a better grasp of the *global* coherence (e.g., the main point or moral of the text, goal of a story, and intention of characters) from the broad context (Graesser et al., 1994). Most questions that are prevalent now require only *local* coherence (e.g., referential relations and thematic roles) with a narrow context. Such questions based on global coherence may be generated by techniques of abstractive text summarization (Rush et al., 2015; Ganesan et al., 2010).

**Annotation issues**: There were questions for which there were disagreements regarding the decision of *nonsense*. For example, some questions can be solved by external knowledge without seeing their context. Thus, we should clarify what a

| Sentence | QA4MRE | MCTest | SQuAD | WDW | MARCO | NewsQA |
|---|---|---|---|---|---|---|
| Number | 1.120 | 1.180 | 1.040 | 1.110 | 1.080 | 1.170 |
| Distance | 1.880 | 0.930 | 0.090 | 0.730 | 0.280 | 0.540 |

Table 6: Average number and distance of sentences that need to be read to answer a question in the RC datasets.

"solvable" or "reasonable" question is in RC. In addition, annotators reported that the prerequisite skills did not easily treat questions whose answer was "none of the above" in QA4MRE. We considered those "no answer" questions difficult in another sense, so our methodology failed to specify them.

**Competence of selecting necessary sentences**: As mentioned in Section 1, our methodology cannot evaluate the competence of selecting sentences that need to be read to answer questions. As a brief analysis, we further investigated sentences in the context of the datasets that were highlighted in the annotation. Analyses were performed in two ways: for each question, we counted the number of required sentences and their distance (see Appendix B for the calculation method). The values of the first row in Table 6 show the average number of required sentences per question for each RC dataset. Although the scores seemed to be approximately level, MCTest required multiple sentences the most frequently. The second row presents the average distance of required sentences. QA4MRE demonstrated the longest distance because readers had to look for clues in long context texts of the dataset. In contrast, SQuAD and MS MARCO showed lower scores: most of their questions seem only to require reading a single sentence to answer. Of course, the scores of distances should depend on the length of the context texts.

## 7 Conclusion

In this study, we adopted evaluation metrics to analyze both the performance of a system and the quality of RC datasets. We assumed two classes—refined *prerequisite skills* and *readability*—and defined evaluation metrics for each. Next, we annotated six existing RC datasets with those defined metrics. Our annotation highlights the characteristics of the datasets and provides a valuable guide for the construction of new datasets and the development of RC systems. For future work, we plan to use the analysis in the present study to construct a system that can be applied to multiple datasets.

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

# A  Sampling Methods for Questions

In this appendix, we explain the method of choosing questions for the annotation.

**QA4MRE** (Sutcliffe et al., 2013): The gold-standard dataset consists of four different topics and four documents for each topic. We randomly selected 100 main and auxiliary questions so that at least one question of each document was included.

**MCTest** (Richardson et al., 2013): This dataset consists of two sets: MC160 and MC500. Their development sets have 80 tasks in total; each includes context texts and four questions. We randomly chose 25 tasks (100 questions) from the development sets.

**SQuAD** (Rajpurkar et al., 2016): This dataset includes some Wikipedia articles from various topics, and those articles are divided into paragraphs. We randomly chose 100 paragraphs from

15 articles and used only one question of each paragraph for the annotation.

**Who-did-What** (WDW) (Onishi et al., 2016): This dataset is constructed from the English Gigaword newswire corpus (v5). Its questions are automatically created from an article that differs from one used for context. In addition, questions that can be solved by a simple baseline method are excluded from the dataset.

**MS MARCO** (MARCO) (Nguyen et al., 2016): A task of this dataset comprises several segments, one question, and its answer. We randomly chose 100 tasks (100 questions) and only used segments whose attribute was *is_selected* = 1 as context.

**NewsQA** (Trischler et al., 2016): we randomly chose questions that satisfied the following conditions: *is_answer_absent* = 0, *is_question_bad* = 0, and *validated_answers* do not include *bad_question* or *none*.

## B  Calculation of Sentence Distance

As mentioned in Section 6, the distance of sentences was calculated as follows. If a question required only one sentence to be read, its distance was zero. If a question required two adjacent sentences to be read, its distance was one. If a question required more than two sentences to be read, its distance was the sum of distances of any two sentences.

