# Peer review of "Evaluation Metrics for Machine Reading Comprehension: Prerequisite Skills and Readability"

_ACL 2017 — decision unknown_

[Official Review · Reviewer 1 · rating 4 · confidence 3]
soundness 4 · originality 3 · clarity 2 · impact 2 · substance 4 · appropriateness 4 · meaningful comparison 3 · presentation format Oral Presentation

- Strengths:

- this article puts two fields together: text readability for humans and
machine comprehension of texts

- Weaknesses:

- The goal of your paper is not entirely clear. I had to read the paper 4 times
and I still do not understand what you are talking about!
- The article is highly ambiguous what it talks about - machine comprehension
or text readability for humans
- you miss important work in the readability field
- Section 2.2. has completely unrelated discussion of theoretical topics.
- I have the feeling that this paper is trying to answer too many questions in
the same time, by this making itself quite weak. Questions such as “does text
readability have impact on RC datasets” should be analyzed separately from
all these prerequisite skills.

- General Discussion:

- The title is a bit ambiguous, it would be good to clarify that you are
referring to machine comprehension of text, and not human reading
comprehension, because “reading comprehension” and “readability”
usually mean that.
- You say that your “dataset analysis suggested that the readability of RC
datasets does not directly affect the question difficulty”, but this depends
on the method/features used for answer detection, e.g. if you use
POS/dependency parse features.
- You need to proofread the English of your paper, there are some important
omissions, like “the question is easy to solve simply look..” on page 1.
- How do you annotate datasets with “metrics”??
- Here you are mixing machine reading comprehension of texts and human reading
comprehension of texts, which, although somewhat similar, are also quite
different, and also large areas.
- “readability of text” is not “difficulty of reading contents”. Check
this:
DuBay, W.H. 2004. The Principles of Readability. Costa Mesa, CA: Impact
information. 
- it would be good if you put more pointers distinguishing your work from
readability of questions for humans, because this article is highly ambiguous.
E.g. on page 1 “These two examples show that the readability of the text does
not necessarily correlate with the difficulty of the questions” you should
add “for machine comprehension”
- Section 3.1. - Again: are you referring to such skills for humans or for
machines? If for machines, why are you citing papers for humans, and how sure
are you they are referring to machines too?
- How many questions the annotators had to annotate? Were the annotators clear
they annotate the questions keeping in mind machines and not people?